# Prenatal Healthy Dietary Patterns Are Associated with Reduced Behavioral Problems of Preschool Children in China: A Latent Class Analysis

**DOI:** 10.3390/ijerph20032214

**Published:** 2023-01-26

**Authors:** Lianjie Dou, Jijun Gu, Ying Pan, Dan Huang, Zhaohui Huang, Huihui Bao, Wanke Wu, Peng Zhu, Fangbiao Tao, Jiahu Hao

**Affiliations:** 1Department of Epidemiology and Health Statistics, School of Public Health, Anhui Medical University, Hefei 230032, China; 2Department of Maternal, Child & Adolescent Health, School of Public Health, Anhui Medical University, Hefei 230032, China; 3Key Laboratory of Population Health Across Life Cycle, Anhui Medical University, Ministry of Education of the People’s Republic of China, Hefei 230032, China; 4NHC Key Laboratory of Study on Abnormal Gametes and Reproductive Tract, Hefei 230032, China; 5Anhui Provincial Key Laboratory of Population Health and Aristogenics/Key Laboratory of Environmental Toxicology of Anhui Higher Education Institutes, Anhui Medical University, Hefei 230032, China; 6Anhui Provincial Center for Women and Child Health, Hefei 230001, China

**Keywords:** dietary patterns, pregnancy, behavioral problems, child, birth cohort

## Abstract

**Highlights:**

Southern dietary pattern was characterized by higher vegetable and fruit intakes.Southern dietary pattern predicted fewer childhood behavioral problems.Latent class analysis was suitable to categorize prenatal food groups intakes.The effect of maternal dietary patterns on child behavior displayed sex differences.

**Abstract:**

The relation between maternal dietary patterns during pregnancy and offspring behavioral problems is less verified. Therefore, we have aimed to assess the relationship between them and have hypothesized that children of mothers with healthy dietary patterns during pregnancy have better behavior. The 1612 mother-child pairs of the China-Anhui Birth Cohort Study (C-ABCS) have been enrolled as the study population. The dietary behaviors of mothers during early and mid-pregnancy have been investigated using a semi-quantitative food frequency questionnaire. Preschool child behavioral problems have been assessed. Clusters of maternal food groups intakes have been identified using latent class analysis, and the association between maternal dietary patterns and child behavioral problems has been subsequently analyzed using logistic regression. Maternal age at inclusion is 26.56 ± 3.51 years. There has been a preponderance of boys (53.3%). Maternal food groups intakes have been classified into four groups: “High-consumed pattern (HCP)”, “Southern dietary pattern (SDP)”, “Northern dietary pattern (NDP)”, and “Low-consumed pattern (LCP)”. The offspring with maternal SDP and NDP have lower emotional symptoms compared to the offspring with maternal LCP in the first trimester (*p* < 0.05). It has been reported to lower conduct problems in children with maternal SDP than the children with maternal LCP in the second trimester (*p* < 0.05). In boys, we have detected associations between first-trimester SDP and lower emotional symptoms (*p* < 0.05) and between second-trimester SDP with decreased peer relationship problems (*p* < 0.05). In girls, total difficulty scores are lower with second-trimester SDP (*p* < 0.05). Maternal SDP in early and mid-pregnancy predicts reduced behavioral problems in preschool children, while maternal HCP and NDP during pregnancy may result in fewer developmental benefits.

## 1. Introduction

The importance of maternal nutrition during pregnancy for fetal brain development has been well documented. Beginning from around 18 days post-fertilization, the embryo undergoes a coordinated process of nerve proliferation and migration, synaptogenesis, myelination, and apoptosis to develop and form the fetal brain [1], but the brain is more vulnerable to nutritional deficiencies at this time. Additionally, the hypothesis of the Developmental Origins of Health and Disease (DOHaD) suggests that during this period of rapid development, the brain becomes more sensitive to the environment, and this is a vulnerable and critical period of perturbation that may predispose the fetus to postnatal neuropsychological disorders [2,3,4].

Human epidemiological evidence has identified an association between maternal nutrient deficiencies during pregnancy and cognitive development of their offspring. Prenatal vitamin A, folic acid, and vitamin D deficiencies are associated with subsequent suboptimal neuropsychological development [4,5,6], such as susceptibility to autism and delayed language development, but several nutrients are not sufficient to assess the nutritional status of pregnant women, and fortunately, birth cohorts on the association of maternal dietary patterns during pregnancy with neuropsychological development of offspring have bridged this gap. Cohort findings have implied that unhealthy dietary patterns during pregnancy are associated with reduced executive function, delayed language development, and lower IQ scores in offspring, with the unhealthy diet including lower Mediterranean diet scores [7,8,9,10,11]. However, the western Mediterranean diet (higher intake of fruits, vegetables, fish, pasta, and rice, and lower intake of meat, sugar, and fat) differs from the eastern dietary pattern (predominantly carbohydrates, vegetables, fruits, pork, etc.). Furthermore, another difference is inland and coastal diets in China because of the higher intake of aquatic products in coastal diets. Therefore, it is necessary to investigate the association between dietary patterns of pregnant women and behavioral problems of offspring in China’s inland. In addition, most studies use principal component analysis (PCA) [9,11,12,13] or cluster analysis [14,15] to classify food groups intakes, but some studies prefer to use latent class analysis (LCA), which is recommended for food intakes to study the effects of mutually exclusive categories [16,17].

Accordingly, we hypothesize that children of mothers with healthy dietary patterns during pregnancy have fewer behavioral problems. This paper aims to classify maternal food group intake into appropriate categories during pregnancy in inner-city China by the LCA method and then analyze the association of dietary patterns with behavioral problems of offspring at the preschool age.

## 2. Methods and Materials

### 2.1. Study Population

This study is based on the China-Anhui Birth Cohort Study (C-ABCS), which has been established in six municipal health institutions between November 2008 and October 2010 with 5084 pregnant women and their offspring recruited. Specific inclusion and exclusion criteria are described in the team’s previously published literature [18]. After excluding maternal loss (202), spontaneous abortions (92), stillbirths, fetal death, induced labor (55), and twin pregnancy (66), 4669 pairs of mothers and singleton live births have been included in the child follow-up cohort. Between April 2014 and April 2015, we have accessed cognitive and behavioral development at early childhood (4.25 ± 0.41 years) using assessment tools that include Strengths and Difficulties Questionnaire (Edition for parents, SDQ), Clancy Autism Behavior Scale (CABS), and Conner’s Abbreviated Symptom Questionnaire (C-ASQ). However, the team consists of several groups, and our group has participated in a survey of the former 1783 mothers, thus obtaining survey data from 1783 mother-child pairs. Among them, 171 mothers have been excluded for no food intakes data, and the data of 1612 mother-child pairs is finally included in the analysis. Figure 1 provides a more visual description.

### 2.2. Measurements

#### 2.2.1. Food Groups Intakes Assessment during Pregnancy

Based on collected literatures and consultation with experts, a semi-quantitative food frequency questionnaire (FFQ) has been composed by selecting food items that represent the dietary intakes of pregnant women in Anhui province, China. The questionnaire is administered at 12.13 ± 3.82 and 30 ± 2.11 gestational weeks, asking about dietary intakes during the first and second trimester. A total of 19 food items have been included, which are rice, wheaten food, vegetables, fruits, beef and mutton, poultry, pork, animal fishery products, eggs, dairy products, beans, nuts, fried foods, pickles, animal innards, and garlic. For each food entry, pregnant women are asked about the frequency of intake in a week, and the options are divided into 5 levels: 1 = no intake, 2 = 1 to 3 times per week, 3 = 4 to 5 times per week, 4 = 6 to 8 times per week, and 5 = more than 9 times per week. The data of food intakes are a skewed distribution and we would regroup it. Referring to the relevant literature [16,17] and the actual distribution of the intake frequency, the criteria for regrouping are as follows. The percentage of non-consumers (option was no intake) is less than 7.5%, and variables are transformed into binary variables: above median and below median. The percentage of non-consumers is higher than 7.5% and lower than 45%, and variables are transformed into triple variables: non-consumed, below median, and above median. The percentage of non-consumers is higher than 45%, and variables are transformed into binary variables: non-consumed and consumed.

#### 2.2.2. Outcomes

The behavioral problems in early childhood are assessed by the SDQ, C-ASQ, and CABS, which is fulfilled by a familiar caregiver and then reviewed by trained investigators.

SDQ refers to children’s behavior and emotions over the previous six months. The scale provides balanced coverage of emotional symptoms, conduct problems, hyperactivity/inattention, peer relationship problems, and prosocial behavior. The former four scales are added together to generate a total difficulties score. The higher the score of total difficulties score, the more serious the objective difficulty is, and the delineation criteria of boundary values refer to the scoring rules [19].

The 10-item C-ASQ is derived from the revised Conners Parent Rating Scale. The widely used scale is used to assess attention-deficit hyperactivity disorder (ADHD) symptoms in children. The options in this scale range from 0 (never) to 3 (frequently) according to the frequency of symptoms. ADHD symptoms have been defined as a total score of ≥15 [20].

CABS is used as a screening tool to identify children with autism. The scale consists of 14 items, with scores of 0, 1, and 2 assigned to “never”, “occasionally”, and “often”, respectively, and a total score of ≥14 is considered positive for potential autism [21].

#### 2.2.3. Covariates

Socio-demographic variables have been investigated in a self-administered maternal and child health record form, including mainly maternal age, education level, place of residence, monthly income, type of work, secondhand smoke exposure, and home renovation at the time of inclusion. We have also extracted children’s birth date, sex of the child, birth weight, and gestational weeks of delivery from hospital birth records at the time of delivery. It should be noted that child age is calculated as the date of examination minus the date of birth, and for preterm infants, age is calculated as the date of testing minus the expected date of delivery. As pregestational body mass index (BMI) = pregestational weight (kg)/maternal height^2^ (m^2^), BMI categories specific for adult Chinese female are assigned as follows: BMI < 18.5 (underweight), BMI  =  18.5–24 (normal), BMI ≥ 24 (obesity or overweight). Maternal depression is assessed by the center for epidemiological survey depression scale (CES-D) [22], with a score above 16 indicating possible depression.

### 2.3. Statistical Analysis

To identify mutually exclusive groupings, we have used LCA to derive dietary patterns. A trivial 1-class model is first fitted in which all individuals belong to the same category, and then 2 to 5-category models are fitted. The optimal model is selected based on BIC and AIC values while considering the same number of categories and reasonable category probabilities in different trimesters to ensure substantial dimensionality reduction in food intakes, ease of model understanding, and further analysis. The names of the clusters are chosen based on the conditional distribution of food intakes. We have four clusters of food intakes, called “High-consumed pattern (HCP)”, “Southern dietary pattern (SDP)”, “Northern dietary pattern (NDP)”, and “Low-consumed pattern (LCP)”.

Afterward, the response probabilities of the potential classes are described. The covariates in the models include mainly maternal age at inclusion, pregestational BMI, maternal education, residential region, monthly income, maternal depression, child gender, and child age at the visit. We also explore differences in the distribution of dietary patterns across covariates using χ^2^ test. Finally, we used a logistic regression model to access the association of dietary patterns with behavioral problems in early childhood. To verify the stability of the regression analysis results, three analytical models with different covariates have been constructed. Model 1 only includes the first trimester and second trimester dietary patterns. Secondarily, model 2 includes covariates such as child gender and age, maternal education, residence, maternal age, pregestational BMI, and monthly income, and model 3 takes into account maternal depression during pregnancy on the basis of model 2. All above analyses have been performed in Mplus 7.4 and SPSS 23.0.

## 3. Results

### 3.1. Demographic Characteristics of the Study Population

The mean ± SD age of pregnant women at inclusion is 26.56 ± 3.51 years, mostly residing in urban areas, and the highest percentage of those report a monthly income of less than 2000 CNY. There is a preponderance of boys (53.3%).

### 3.2. Latent Profiles of Food Intakes

Based on the lower BIC and AIC parameters in different classes of the LCA models, we have chosen to classify the first trimester and second trimester food groups intakes into four classes (Table 1). The class probability in the first and second trimesters from HCP, SDP, NDP, to LCP shows as follows: the first trimester: 0.20, 0.39, 0.18, 0.23; the second trimester: 0.19, 0.44, 0.16, 0.21. 

### 3.3. Probabilities of Food Groups Intakes

Table 2 and Table 3 present the conditional distribution of food groups intakes in the first and second trimesters, making the latent classes of food intakes clearer in differentiating and labeling the clusters. Cluster 1 has the higher probability of food groups intakes in the questionnaire, so we call this cluster “HCP”. Contrary to cluster 1, cluster 2 has the lowest intakes of food groups under investigation, so we have termed cluster 2 “LCP”. Subjects in cluster 3, “SDP”, report high consumption of rice, vegetables, and fruits, and it is in line with the dietary habits of people in southern China. But compared to the cluster 3, cluster 4 has the higher consumption of wheaten food and meat, and it is similar to that of people in northern China.

### 3.4. Distribution of Maternal Diet Patterns

Table 4 reports the differences in the distribution of dietary patterns in the first and second trimester across different demographic characteristics. Higher maternal age, monthly income, place of residence, and education level are associated with significant differences of dietary pattern distribution in both trimesters (*p* < 0.05). Similarly, the distribution in the first trimester is different among different pre-conceptional BMI groups. In the second trimester, significant differences are observed between depressed and non-depressed mothers in the dietary pattern distribution. Nevertheless, no association is observed between child gender and maternal dietary patterns. 

### 3.5. Associations of Maternal Dietary Patterns with Child Behavioral Problems

The results of behavioral tests in children are shown in Table 5. The SDQ examination data indicate that 121 (7.8%) of children have presented with behavioral problems, with the highest rate of hyperactive behavior (8.6%) and the lowest rate of peer interaction (2.6%). Meanwhile, the C-ASQ test results suggest 158 (9.8%) children with possible hyperactivity problems, and CABS test results indicate 122 (7.6%) children with a potential autistic behavioral problems.

After adjusting for child gender and age, maternal sociodemographic characteristics, and depression, the offspring with maternal SDP and NDP have lower emotional symptoms compared to the offspring with maternal LCP in the first trimester [OR_SDP_ (95%CI), 0.4 (0.28, 0.87); OR_NDP_ (95%CI), 0.47 (0.24, 0.95)] (Table 6 and Table 7). Then, it has been reported to lower conduct problems in children with maternal SDP than the children with maternal LCP in the second trimester [OR (95%CI): 0.55 (0.34, 0.9)]. Results in sex-stratified analyses are slightly different. The relation of emotional symptoms and maternal SDP in early pregnancy is only observed in boys [OR (95%CI):0.35 (0.13, 0.95)] (Table 8). In addition, the associations are noted between maternal SDP in the second trimester and decreased peer relationship problems in boys [hyperactivity, OR (95%CI): 0.35 (0.15, 0.85); Peer relationship problems, OR (95%CI): 0.27 (0.09, 0.79)], and between maternal SDP in the second trimester and reduced hyperactivity [OR (95%CI); 0.35 (0.15, 0.85)]. Inversely, the association between conduct problems and maternal dietary patterns during mid-pregnancy is found only in girls. Maternal SDP in the second trimester predicts lower total difficulties score in girls [OR (95%CI): 0.44 (0.2, 0.95)]. No association was observed between dietary patterns during pregnancy and ADHD and autism.

## 4. Discussion

In this birth cohort study in inland China, we have observed that maternal SDP (characterized by higher vegetable and fruit intakes), NDP (characterized by higher meat intakes), and HCP (characterized by high food groups intakes) during early pregnancy are associated with lower incidence of emotional symptoms in preschool-age children compared with LCP (characterized by lower food groups intakes), and maternal SDP at mid-pregnancy is associated with reduced conduct problems in children. In addition, we have detected associations between maternal SDP in early pregnancy and lower emotional symptoms, and between maternal SDP in mid-pregnancy with decreased peer relationship problems in boys. In girls, total difficulty scores are lower with maternal SDP in mid-pregnancy. Overall, these findings supported our hypothesis, and maternal SDP in both early and mid-pregnancy may predict fewer childhood behavioral problems, but maternal HCP and NDP during pregnancy unlikely result in reduced behavioral problems.

Several birth cohort studies have reported an association between unhealthy maternal dietary patterns during pregnancy and decreased behavioral problems in the offspring. A few of these studies have had several dietary patterns that are slightly similar to the dietary patterns in this paper. The Avon Longitudinal Study of Parents and Children in the United Kingdom shows that 8-year-old children of mothers in the “vegetables and fruits” dietary cluster have higher IQs, while an unhealthy maternal diet during pregnancy (processed and junk foods) is associated with lower cognitive function in 7- and 8-year-old children [23]. The EDEN mother-child cohort in France reports a positive association between maternal “low health diet (characterized by low intake of fruits, vegetables, fish and whole grains)” and “high Western diet (processed foods and snacks)” during pregnancy and a high symptomatic ADHD-attention trajectory in children aged 3 to 8 years [7]. The Generation R Study and The Norwegian Mother and Child Cohort Study (MoBa) find similar conclusions: unhealthy dietary patterns during pregnancy are positively associated with externalizing behavior (inattention, aggression) in the offspring [8,11]. A US cohort has reported that maternal intake of a higher quality diet during pregnancy (higher Mediterranean diet score or Alternative Health Diet Index) is associated with better visuospatial skills in offspring at 3.2 years of age and better intellectual and executive functioning in offspring at 7.7 years of age [10]. Similarly, a birth cohort study in the coastal city of Jiangsu, China, identifies a high intake of dietary fiber and high-quality protein (aquatic products, fresh vegetables) during mid and late pregnancy as predictive of better gross motor and receptive communication development in 1-year-old children [9]. However, not all studies have found an association between diet during pregnancy and behavior problems in offspring, and data from The Southampton Women’s Survey does not reveal an association between vegetarian consumption during pregnancy and poorer cognitive development in children aged 6–7 years [24]. In addition, two meta-analyses have implied that higher maternal diet quality is associated with a lower risk of poorer cognitive development in offspring [5,25]. The above studies confirm the plausibility of the findings of this study.

This paper represents the children of mothers with a southern dietary pattern of higher vegetable and fruit intake that have reduced behavioral problems compared to mothers with a low intake diet. Vegetables and fruits provide the macronutrients (vitamin A, vitamin C, carotenoids, and small amounts of B vitamins) and key minerals (calcium, magnesium, potassium, and iron) that the fetal brain needs to develop in utero. Moreover, low levels of food intake imply inadequate nutrient intakes. Sub-optimal macronutrient balance and micronutrient deficiencies can lead to poor maternal body composition and metabolism, which in turn can affect maternal health and lead to intrauterine programming of the fetus, altering fetal brain morphogenesis, brain neurochemistry, and neurophysiology long-term metabolic and cognitive health consequences [2,26,27].

This study has several strengths. To our knowledge, this study is one of the few studies to use the LCA method to categorize food groups intakes during pregnancy and then explore the association with reduced behavioral problems of offspring in early childhood. LCA is applicable to a wide range of variable types (categorical and continuous variables) and provides higher classification accuracy, for it is based on probabilistic mixture modeling. It is also suitable for missing data [28,29]. This study is an inland Chinese prospective birth cohort study that investigates maternal diets at both early and mid-pregnancy visits. However, this study also has several limitations. Firstly, the main drawback of this study is that dietary patterns are assessed based on frequency of intake rather than the actual amount consumed. Secondly, the data is part of the cohort study data and not all data are available. Thirdly, it is unable to investigate the food intakes in late pregnancy and could not assess dietary information throughout pregnancy. Fourthly, the evaluation results of children’s behavioral problems, especially ASD and ADHD, are obtained through questionnaires and have not been diagnosed by special clinicians, which makes the evaluation results reliable. In addition, C-ABCS is not a national cohort and the dietary patterns in the paper are only representative of the diet of people living in central China, which makes extrapolation of the results of this study limited. Finally, although the design of the food frequency questionnaire has been discussed and modified several times, it could not cover all types of foods consumed by pregnant women, such as the lack of root and tuber crops. The deficiencies of the study do not deny its value.

## 5. Conclusions

This study finds that the maternal southern dietary pattern (characterized by higher vegetable and fruit intakes) is predictive of reduced behavior problems in early childhood, suggesting that health care providers should strengthen maternal knowledge about nutrition during pregnancy, especially to ensure fruit and vegetable intake.

## Figures and Tables

**Figure 1 ijerph-20-02214-f001:**
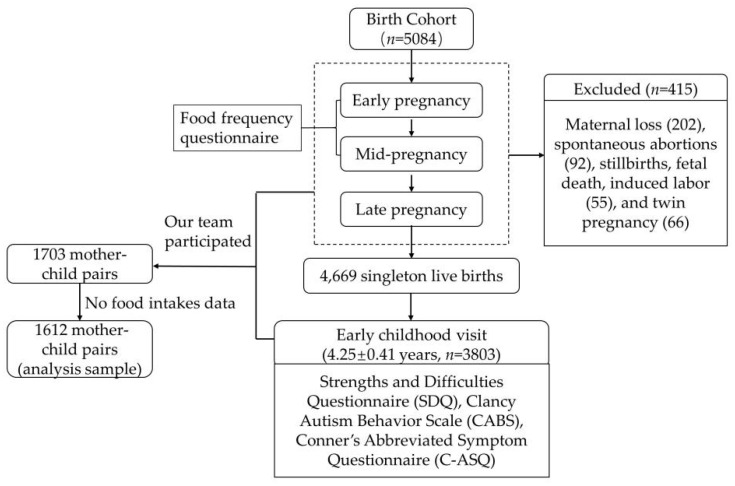
Flow-diagram of the cohort participants.

**Table 1 ijerph-20-02214-t001:** Demographic characteristics of analysis samples, *n* (%).

Characteristics	*n* (%)
Monthly income per capita (CNY)	
<2000	950 (59)
2000–4000	541 (33.7)
≥4000	118 (7.3)
Educational attainment	
Junior middle school and below	390 (24.2)
High school	447 (27.8)
Junior college	406 (25.3)
Undergraduate and above	366 (22.7)
Place of residence	
Urban	1393 (86.5)
Not in Urban	217 (13.5)
Maternal age (Year)	
19–24	469 (29.1)
25–29	887 (55)
≥30	256 (15.9)
Preconceptional BMI	
Underweight	397 (24.7)
Normal	1117 (69.4)
Obesity or overweight	95 (5.9)
Maternal depression	
No	1559 (96.8)
Yes	52 (3.2)
Sex of child	
Boy	860 (53.3)
Girl	752 (46.7)
Preterm birth (<37 gestational weeks)	
No	1571 (97.5)
Yes	41 (2.5)
Birth weight	
No	1591 (98.7)
Yes	21 (1.3)
Mode of delivery	
Cesarean delivery	565 (35.0)
Vaginal delivery	1047 (65.0)

Abbreviations: CNY, Chinese Yuan.

**Table 2 ijerph-20-02214-t002:** Fit statistics for a series of latent class analysis models of prenatal food groups intakes.

Class	K	Log (L)	AIC	BIC	aBIC	Entropy	LMR (*p*)	BLRT (*p*)	Class Probability
First trimester									
1 class	64	−32,405.62	64,939.24	65,283.90	65,080.58	1			
2 classes	129	−30,919.11	62,096.22	62,790.91	62,381.10	0.788	<0.001	<0.001	0.49/0.51
3 classes	194	−30,445.59	61,279.18	62,323.91	61,707.61	0.791	<0.001	<0.001	0.22/0.53/0.25
4 classes	259	−30,154.61	60,827.21	62,221.99	61,399.19	0.780	<0.001	0.028	0.23/0.39/0.18/0.20
5 classes	324	−29,959.70	60,567.40	62,312.21	61,282.92	0.798	0.436	>0.999	0.07/0.39/0.22/0.17/0.14
Second trimester									
1 class	22	−20,122.03	40,288.06	40,406.53	40,336.64				
2 classes	45	−19,143.15	38,376.31	38,618.64	38,475.69	0.715	<0.001	<0.001	0.50/0.50
3 classes	68	−18,811.03	37,758.05	38,124.25	37,908.22	0.735	<0.001	<0.001	0.29/0.44/0.28
4 classes	91	−18,657.21	37,496.41	37,986.47	37,697.38	0.746	0.013	0.014	0.21/0.16/0.19/0.44
5 classes	114	−18,576.52	37,381.04	37,994.96	37,632.80	0.719	0.280	0.283	0.09/0.26/0.19/0.15/0.32

Abbreviations: AIC, Akaike information criterion; aBIC, adjusted Bayesian information criterion; BIC, Bayesian information criterion; BLRT (*p*): *p*-Value for the bootstrapped likelihood ratio test; LMR (*p*): *p*-Value for the adjusted Lo-Mendell-Rubin-test; Log (L): Log-likelihood.

**Table 3 ijerph-20-02214-t003:** Probabilities of the first-trimester food groups intakes derived from LCA, %.

Food Groups Intakes	HCP	SDP	NDP	LCP
Rice				
Below median	0.131	0.198	0.67	0.716
Above median	0.869	0.802	0.33	0.284
Wheaten food				
Below median	0.232	0.46	0.358	0.618
Above median	0.768	0.54	0.642	0.382
Vegetables				
Below median	0.144	0.312	0.825	0.937
Above median	0.856	0.688	0.175	0.063
Fruits				
Below median	0.112	0.145	0.382	0.741
Above median	0.888	0.855	0.618	0.259
Beef and mutton				
Not consumed	0.229	0.537	0.239	0.672
Below median	0.459	0.414	0.54	0.3
Above median	0.312	0.049	0.221	0.028
Pork				
Not consumed	0.019	0.077	0.011	0.252
Below median	0.099	0.477	0.351	0.625
Above median	0.883	0.446	0.639	0.123
Poultry				
Not consumed	0.024	0.219	0.035	0.461
Below median	0.272	0.666	0.582	0.514
Above median	0.704	0.115	0.382	0.025
Animal fishery products				
Not consumed	0.011	0.093	0.028	0.297
Below median	0.235	0.691	0.488	0.662
Above median	0.755	0.216	0.484	0.041
Eggs				
Below median	0.072	0.357	0.189	0.785
Above median	0.928	0.643	0.811	0.215
Dairy products				
Not consumed	0.027	0.15	0.021	0.3
Below median	0.256	0.447	0.628	0.618
Above median	0.717	0.403	0.351	0.082
Beans				
Not consumed	0	0.101	0.011	0.252
Below median	0.109	0.524	0.411	0.662
Above median	0.891	0.375	0.579	0.085
Nuts				
Not consumed	0.077	0.209	0.067	0.445
Below median	0.227	0.431	0.516	0.479
Above median	0.696	0.359	0.418	0.076
Fried foods				
Not consumed	0.405	0.561	0.274	0.621
Consumed	0.595	0.439	0.726	0.379
Pickles				
Not consumed	0.189	0.231	0.112	0.338
Below median	0.453	0.482	0.632	0.527
Above median	0.357	0.287	0.256	0.136
Animal innards				
Not consumed	0.283	0.669	0.295	0.785
Consumed	0.717	0.331	0.705	0.215
Garlic				
Not consumed	0.288	0.466	0.302	0.644
Below median	0.429	0.446	0.611	0.338
Above median	0.283	0.088	0.088	0.019

Abbreviations: LCA, latent class analysis; LCP, Low-consumed pattern; SDP, Southern dietary pattern; NDP, Northern dietary pattern.

**Table 4 ijerph-20-02214-t004:** Probabilities of the second-trimester food groups intakes derived from LCA, %.

Food Groups Intakes	HCP	SDP	NDP	LCP
Rice				
Below median	0.123	0.07	0.578	0.558
Above median	0.877	0.93	0.422	0.442
Wheaten food				
Below median	0.17	0.482	0.205	0.601
Above median	0.83	0.518	0.795	0.399
Vegetables				
Below median	0.07	0.098	0.709	0.919
Above median	0.93	0.902	0.291	0.081
Fruit				
Below median	0.14	0.219	0.95	0.903
Above median	0.86	0.781	0.05	0.097
Beef and mutton				
Not consumed	0.146	0.459	0.194	0.523
Below median	0.485	0.493	0.36	0.438
Above median	0.368	0.048	0.446	0.039
Pork				
Below median	0.041	0.298	0.295	0.679
Above median	0.959	0.702	0.705	0.321
Poultry				
Not consumed	0.003	0.091	0.019	0.185
Below median	0.108	0.641	0.384	0.718
Above median	0.889	0.268	0.597	0.097
Animal fishery products				
Below median	0.085	0.592	0.314	0.896
Above median	0.915	0.408	0.686	0.104
Eggs				
Below median	0.12	0.401	0.395	0.795
Above median	0.88	0.599	0.605	0.205
Dairy products				
Below median	0.096	0.351	0.395	0.766
Above median	0.904	0.649	0.605	0.234
Beans				
Below median	0.026	0.49	0.26	0.818
Above median	0.974	0.51	0.74	0.182
Nuts				
Not consumed	0.044	0.193	0.07	0.338
Below median	0.228	0.509	0.523	0.565
Above median	0.728	0.298	0.407	0.097
Fried foods				
Not consumed	0.289	0.543	0.326	0.581
Consumed	0.711	0.457	0.674	0.419
Pickles				
Not consumed	0.281	0.369	0.217	0.38
Below median	0.509	0.496	0.64	0.555
Above median	0.211	0.135	0.143	0.065
Animal innards				
Not consumed	0.14	0.433	0.244	0.552
Below median	0.556	0.547	0.632	0.448
Above median	0.304	0.02	0.124	0
Garlic				
Not consumed	0.225	0.365	0.283	0.464
Below median	0.43	0.499	0.558	0.487
Above median	0.345	0.136	0.159	0.049

Abbreviations: LCA, latent class analysis; LCP, Low-consumed pattern; SDP, Southern dietary pattern; NDP, Northern dietary pattern.

**Table 5 ijerph-20-02214-t005:** Distribution of sociodemographic characteristics by prenatal maternal dietary patterns.

Characteristics	First Trimester				*p*	Second Trimester				*p*
	LCP	SDP	NDP	HCP		LCP	SDP	NDP	HCP	
Monthly income per capita (CNY)										
<2000	217 (68.7)	354 (55.8)	164 (57.7)	215 (57.3)	0.007	216 (70.4)	412 (58.6)	142 (55.3)	180 (52.6)	<0.001
2000–4000	77 (24.4)	236 (37.2)	96 (33.8)	132 (35.2)		80 (26.1)	245 (34.9)	94 (36.6)	122 (35.7)	
≥4000	22 (7)	44 (6.9)	24 (8.5)	28 (7.5)		11 (3.6)	46 (6.5)	21 (8.2)	40 (11.7)	
Educational attainment										
Junior middle school and below	117 (37)	163 (25.7)	54 (19)	56 (14.9)	<0.001	136 (44.3)	156 (22.2)	54 (21)	44 (12.9)	<0.001
High school	91 (28.8)	160 (25.2)	91 (32)	105 (28)		80 (26.1)	205 (29.2)	75 (29.2)	87 (25.4)	
Junior college	67 (21.2)	161 (25.4)	83 (29.2)	95 (25.3)		51 (16.6)	183 (26)	67 (26.1)	105 (30.7)	
Undergraduate and above	41 (13)	150 (23.7)	56 (19.7)	119 (31.7)		40 (13)	159 (22.6)	61 (23.7)	106 (31)	
Place of residence										
Urban	264 (83.3)	540 (85.2)	250 (88)	339 (90.4)	0.026	247 (80.2)	612 (87.1)	222 (86.4)	312 (91.2)	0.001
Not in Urban	53 (16.7)	94 (14.8)	34 (12)	36 (9.6)		61 (19.8)	91 (12.9)	35 (13.6)	30 (8.8)	
Maternal age (Year)										
19–24	122 (38.5)	178 (28)	84 (29.5)	85 (22.7)	0.001	113 (36.7)	199 (28.3)	63 (24.4)	94 (27.5)	0.042
25–29	149 (47)	358 (56.4)	159 (55.8)	221 (58.9)		148 (48.1)	395 (56.1)	148 (57.4)	196 (57.3)	
≥30	46 (14.5)	99 (15.6)	42 (14.7)	69 (18.4)		47 (15.3)	110 (15.6)	47 (18.2)	52 (15.2)	
Preconceptional BMI										
Underweight	78 (24.7)	140 (22.1)	69 (24.3)	110 (29.3)	0.039	77 (25.1)	174 (24.8)	60 (23.3)	86 (25.1)	0.714
Normal	221 (69.9)	465 (73.3)	194 (68.3)	237 (63.2)		209 (68.1)	496 (70.6)	180 (70)	232 (67.8)	
Obesity or overweight	17 (5.4)	29 (4.6)	21 (7.4)	28 (7.5)		21 (6.8)	33 (4.7)	17 (6.6)	24 (7)	
Maternal depression										
No	300 (94.9)	621 (97.8)	275 (96.5)	363 (96.8)	0.132	293 (95.1)	689 (98)	244 (94.6)	333 (97.4)	0.015
Yes	16 (5.1)	14 (2.2)	10 (3.5)	12 (3.2)		15 (4.9)	14 (2)	14 (5.4)	9 (2.6)	
Sex of child										
Boy	162 (51.1)	337 (53.1)	151 (53)	210 (56)	0.629	163 (52.9)	375 (53.3)	148 (57.4)	174 (50.9)	0.469
Girl	155 (48.9)	298 (46.9)	134 (47)	165 (44)		145 (47.1)	329 (46.7)	110 (42.6)	168 (49.1)	

Abbreviations: LCP, Low-consumed pattern; SDP, Southern dietary pattern; NDP, Northern dietary pattern, CNY, Chinese Yuan. Note: The above analysis were completed by chi-square test.

**Table 6 ijerph-20-02214-t006:** Child behavioral outcomes.

Behavioral Outcomes	*n*	Mean ± SD	M (*P*_25_, *P*_75_)	Detected, *n* (%)
Emotional symptoms	1549	1.77 ± 1.55	1 (1, 3)	96 (6.2)
Conduct problems	1549	1.59 ± 1.22	1 (1, 2)	122 (7.9)
Hyperactivity/inattention	1549	4.27 ± 2.17	4 (3, 6)	133 (8.6)
Peer relationship problems	1549	2.41 ± 1.52	2 (1, 3)	41 (2.6)
Total difficulties score	1549	10.04 ± 4.28	10 (7, 13)	121 (7.8)
Prosocial behavior	1549	6.76 ± 1.96	7 (5, 8)	171 (11)
C-ASQ	1599	7.71 ± 4.76	7 (4, 10)	158 (9.8)
CABS	1557	7.19 ± 4.21	7 (4, 10)	122 (7.6)

Abbreviations: CABS, Clancy Autism Behavior Scale; C-ASQ, Conner’s Abbreviated Symptom Questionnaire.

**Table 7 ijerph-20-02214-t007:** Associations of prenatal dietary maternal patterns with child behavioral problems at preschool age.

Behavioral Outcomes	Model 1 ^a^		Model 2 ^b^		Model 3 ^c^	
	OR (95%CI)	*p*	OR (95%CI)	*p*	OR (95%CI)	*p*
Emotional symptoms						
LCP in first trimester	1		1		1	
SDP in first trimester	0.46 (0.26, 0.8)	0.006	0.49 (0.28, 0.85)	0.012	0.49 (0.28, 0.87)	0.014
NDP in first trimester	0.46 (0.23, 0.91)	0.026	0.48 (0.24, 0.95)	0.036	0.47 (0.24, 0.95)	0.036
HCP in first trimester	0.63 (0.34, 1.18)	0.149	0.73 (0.38, 1.37)	0.324	0.73 (0.38, 1.39)	0.339
LCP in second trimester	1		1		1	
SDP in second trimester	0.82 (0.45, 1.48)	0.499	0.87 (0.47, 1.59)	0.647	0.92 (0.5, 1.71)	0.802
NDP in second trimester	1.33 (0.68, 2.62)	0.406	1.52 (0.76, 3.06)	0.241	1.51 (0.74, 3.05)	0.255
HCP in second trimester	1.29 (0.65, 2.56)	0.466	1.38 (0.67, 2.81)	0.38	1.43 (0.69, 2.94)	0.337
Conduct problems						
LCP in first trimester	1		1		1	
SDP in first trimester	1.05 (0.63, 1.73)	0.86	1.1 (0.66, 1.83)	0.709	1.11 (0.67, 1.85)	0.691
NDP in first trimester	1.17 (0.65, 2.1)	0.61	1.19 (0.66, 2.16)	0.565	1.19 (0.66, 2.16)	0.562
HCP in first trimester	0.76 (0.39, 1.45)	0.399	0.83 (0.43, 1.62)	0.59	0.84 (0.43, 1.62)	0.597
LCP in second trimester	1		1		1	
SDP in second trimester	0.55 (0.34, 0.88)	0.014	0.54 (0.33, 0.89)	0.014	0.55 (0.34, 0.9)	0.017
NDP in second trimester	0.58 (0.32, 1.07)	0.08	0.59 (0.32, 1.1)	0.097	0.59 (0.31, 1.09)	0.093
HCP in second trimester	0.58 (0.32, 1.07)	0.079	0.57 (0.3, 1.06)	0.076	0.57 (0.31, 1.07)	0.08
Hyperactivity/inattention						
LCP in first trimester	1		1		1	
SDP in first trimester	0.84 (0.53, 1.35)	0.477	0.89 (0.55, 1.44)	0.641	0.89 (0.55, 1.44)	0.634
NDP in first trimester	0.59 (0.32, 1.1)	0.099	0.63 (0.34, 1.19)	0.157	0.63 (0.33, 1.19)	0.155
HCP in first trimester	0.86 (0.49, 1.51)	0.598	0.94 (0.53, 1.68)	0.836	0.94 (0.52, 1.68)	0.826
LCP in second trimester	1		1		1	
SDP in second trimester	0.84 (0.52, 1.35)	0.465	0.97 (0.6, 1.6)	0.917	0.98 (0.6, 1.61)	0.942
NDP in second trimester	0.81 (0.43, 1.5)	0.495	0.95 (0.5, 1.78)	0.865	0.95 (0.5, 1.79)	0.867
HCP in second trimester	0.84 (0.46, 1.52)	0.562	1.11 (0.6, 2.06)	0.745	1.12 (0.6, 2.07)	0.732
Peer relationships problem						
LCP in first trimester	1		1		1	
SDP in first trimester	1.43 (0.61, 3.38)	0.416	1.4 (0.59, 3.36)	0.446	1.39 (0.58, 3.33)	0.462
NDP in first trimester	0.97 (0.32, 2.9)	0.956	0.95 (0.31, 2.89)	0.931	0.94 (0.31, 2.87)	0.918
HCP in first trimester	1.17 (0.4, 3.47)	0.773	1.11 (0.37, 3.33)	0.857	1.1 (0.37, 3.3)	0.869
LCP in second trimester	1		1		1	
SDP in second trimester	0.46 (0.21, 1.01)	0.054	0.47 (0.21, 1.05)	0.064	0.47 (0.21, 1.04)	0.064
NDP in second trimester	0.55 (0.2, 1.51)	0.244	0.56 (0.2, 1.57)	0.268	0.56 (0.2, 1.58)	0.272
HCP in second trimester	0.46 (0.17, 1.27)	0.133	0.49 (0.17, 1.38)	0.175	0.48 (0.17, 1.38)	0.174
Total difficulties score						
LCP in first trimester	1		1		1	
SDP in first trimester	0.61 (0.37, 1.01)	0.052	0.67 (0.4, 1.12)	0.123	0.7 (0.42, 1.16)	0.167
NDP in first trimester	0.8 (0.45, 1.42)	0.438	0.87 (0.48, 1.55)	0.626	0.88 (0.49, 1.59)	0.677
HCP in first trimester	0.88 (0.49, 1.57)	0.662	1.03 (0.57, 1.88)	0.918	1.06 (0.58, 1.95)	0.843
LCP in second trimester	1		1		1	
SDP in second trimester	0.58 (0.35, 0.95)	0.029	0.63 (0.38, 1.04)	0.071	0.66 (0.39, 1.09)	0.106
NDP in second trimester	0.91 (0.51, 1.61)	0.744	1.03 (0.57, 1.86)	0.922	1.01 (0.56, 1.83)	0.979
HCP in second trimester	0.53 (0.28, 1)	0.049	0.59 (0.3, 1.13)	0.111	0.59 (0.3, 1.15)	0.124
Prosocial behavior						
LCP in first trimester	1		1		1	
SDP in first trimester	0.94 (0.61, 1.45)	0.782	0.91 (0.59, 1.41)	0.67	0.94 (0.6, 1.45)	0.765
NDP in first trimester	0.84 (0.5, 1.43)	0.53	0.85 (0.5, 1.46)	0.561	0.87 (0.51, 1.49)	0.606
HCP in first trimester	0.89 (0.53, 1.5)	0.667	0.88 (0.52, 1.5)	0.64	0.89 (0.52, 1.52)	0.675
LCP in second trimester	1		1		1	
SDP in second trimester	0.88 (0.57, 1.34)	0.542	0.89 (0.58, 1.39)	0.618	0.92 (0.59, 1.43)	0.715
NDP in second trimester	0.9 (0.53, 1.54)	0.704	0.88 (0.51, 1.52)	0.653	0.87 (0.5, 1.51)	0.621
HCP in second trimester	0.69 (0.4, 1.2)	0.185	0.73 (0.41, 1.28)	0.27	0.74 (0.42, 1.31)	0.301
C-ASQ						
LCP in first trimester	1		1		1	
SDP in first trimester	0.75 (0.48, 1.17)	0.21	0.78 (0.5, 1.23)	0.291	0.78 (0.49, 1.23)	0.284
NDP in first trimester	0.67 (0.38, 1.15)	0.146	0.69 (0.39, 1.21)	0.192	0.69 (0.39, 1.2)	0.188
HCP in first trimester	0.81 (0.48, 1.37)	0.426	0.87 (0.51, 1.5)	0.618	0.87 (0.51, 1.49)	0.606
LCP in second trimester	1		1		1	
SDP in second trimester	0.76 (0.48, 1.2)	0.239	0.81 (0.5, 1.3)	0.377	0.81 (0.51, 1.31)	0.396
NDP in second trimester	1.14 (0.66, 1.96)	0.638	1.25 (0.72, 2.19)	0.427	1.26 (0.72, 2.19)	0.425
HCP in second trimester	1.02 (0.59, 1.77)	0.938	1.22 (0.69, 2.16)	0.495	1.23 (0.69, 2.17)	0.484
CABS						
LCP in first trimester	1		1		1	
SDP in first trimester	0.79 (0.48, 1.3)	0.36	0.8 (0.48, 1.33)	0.393	0.81 (0.49, 1.35)	0.42
NDP in first trimester	0.8 (0.44, 1.46)	0.468	0.86 (0.47, 1.58)	0.627	0.86 (0.47, 1.58)	0.621
HCP in first trimester	0.71 (0.38, 1.3)	0.264	0.75 (0.4, 1.39)	0.362	0.75 (0.41, 1.4)	0.37
LCP in second trimester	1		1		1	
SDP in second trimester	0.64 (0.39, 1.06)	0.082	0.71 (0.42, 1.19)	0.188	0.72 (0.43, 1.22)	0.22
NDP in second trimester	0.93 (0.51, 1.68)	0.803	1.02 (0.55, 1.88)	0.955	1.01 (0.55, 1.87)	0.977
HCP in second trimester	0.96 (0.53, 1.73)	0.878	1.13 (0.61, 2.1)	0.692	1.15 (0.62, 2.14)	0.659

Abbreviations: LCP, Low-consumed pattern; SDP, Southern dietary pattern; NDP, Northern dietary pattern, CABS, Clancy Autism Behavior Scale; C-ASQ, Conner’s Abbreviated Symptom Questionnaire. a: Model 1 includes maternal dietary patterns in first and second trimester; b: Model 2 includes covariates such as child gender and age, maternal education, residence, maternal age, pregestational BMI, and monthly income; c: Model 3 takes into account maternal depression during pregnancy on the basis of model 2.

**Table 8 ijerph-20-02214-t008:** Associations of prenatal maternal dietary patterns with the behavioral problems in preschool-age boys and girls.

Behavioral Outcomes	Boys ^a^		Girls ^a^	
	OR (95%CI)	*p*	OR (95%CI)	*p*
Emotional symptoms				
LCP in first trimester	1		1	
SDP in first trimester	0.35 (0.13, 0.95)	0.039	0.57 (0.27, 1.18)	0.127
NDP in first trimester	0.42 (0.14, 1.31)	0.135	0.51 (0.21, 1.27)	0.148
HCP in first trimester	0.69 (0.26, 1.84)	0.459	0.69 (0.28, 1.69)	0.416
LCP in second trimester	1		1	
SDP in second trimester	0.93 (0.32, 2.73)	0.899	0.97 (0.45, 2.08)	0.935
NDP in second trimester	1.5 (0.48, 4.72)	0.489	1.36 (0.53, 3.5)	0.529
HCP in second trimester	1.55 (0.45, 5.36)	0.49	1.49 (0.6, 3.74)	0.391
Conduct problems				
LCP in first trimester	1		1	
SDP in first trimester	1.32 (0.62, 2.81)	0.472	1.03 (0.5, 2.1)	0.944
NDP in first trimester	1.03 (0.42, 2.54)	0.946	1.41 (0.62, 3.17)	0.414
HCP in first trimester	1.03 (0.41, 2.58)	0.958	0.73 (0.27, 1.97)	0.539
LCP in second trimester	1		1	
SDP in second trimester	0.63 (0.3, 1.33)	0.227	0.46 (0.24, 0.91)	0.024
NDP in second trimester	0.99 (0.41, 2.38)	0.987	0.33 (0.12, 0.88)	0.027
HCP in second trimester	0.81 (0.32, 2.05)	0.656	0.4 (0.17, 0.96)	0.04
Hyperactivity/inattention				
LCP in first trimester	1		1	
SDP in first trimester	0.76 (0.41, 1.39)	0.374	1.21 (0.52, 2.78)	0.66
NDP in first trimester	0.35 (0.15, 0.85)	0.02	1.49 (0.55, 4.02)	0.431
HCP in first trimester	0.79 (0.39, 1.6)	0.511	1.17 (0.4, 3.43)	0.779
LCP in second trimester	1		1	
SDP in second trimester	1.11 (0.58, 2.14)	0.757	0.84 (0.38, 1.85)	0.67
NDP in second trimester	1.18 (0.53, 2.66)	0.682	0.68 (0.24, 1.98)	0.483
HCP in second trimester	1.76 (0.8, 3.86)	0.158	0.54 (0.18, 1.64)	0.278
Peer relationships problem				
LCP in first trimester	1		1	
SDP in first trimester	0.91 (0.3, 2.74)	0.87	3.3 (0.66, 16.59)	0.147
NDP in first trimester	0.38 (0.07, 2.03)	0.257	3.29 (0.54, 20.06)	0.197
HCP in first trimester	1.24 (0.35, 4.36)	0.738	/ ^b^	
LCP in second trimester	1		1	
SDP in second trimester	0.27 (0.09, 0.79)	0.017	1.12 (0.31, 4.11)	0.86
NDP in second trimester	0.43 (0.12, 1.57)	0.201	0.77 (0.13, 4.7)	0.779
HCP in second trimester	0.45 (0.13, 1.59)	0.216	0.4 (0.04, 3.92)	0.43
Total difficulties score				
LCP in first trimester	1		1	
SDP in first trimester	0.8 (0.4, 1.61)	0.528	0.55 (0.25, 1.23)	0.146
NDP in first trimester	0.54 (0.22, 1.35)	0.189	1.23 (0.54, 2.81)	0.626
HCP in first trimester	1.02 (0.45, 2.29)	0.962	1.03 (0.4, 2.62)	0.959
LCP in second trimester	1		1	
SDP in second trimester	0.88 (0.42, 1.81)	0.723	0.44 (0.2, 0.95)	0.036
NDP in second trimester	1.15 (0.49, 2.71)	0.757	0.92 (0.39, 2.16)	0.85
HCP in second trimester	0.63 (0.23, 1.7)	0.363	0.57 (0.23, 1.44)	0.233
Prosocial behavior				
LCP in first trimester	1		1	
SDP in first trimester	1.03 (0.58, 1.82)	0.918	0.77 (0.37, 1.6)	0.488
NDP in first trimester	0.92 (0.47, 1.81)	0.802	0.88 (0.35, 2.18)	0.779
HCP in first trimester	0.82 (0.42, 1.61)	0.559	1.12 (0.46, 2.76)	0.799
LCP in second trimester	1		1	
SDP in second trimester	0.84 (0.48, 1.48)	0.551	1.01 (0.48, 2.12)	0.984
NDP in second trimester	0.85 (0.43, 1.69)	0.65	0.84 (0.33, 2.13)	0.706
HCP in second trimester	0.95 (0.48, 1.89)	0.883	0.38 (0.13, 1.16)	0.09
C-ASQ				
LCP in first trimester	1		1	
SDP in first trimester	0.73 (0.41, 1.31)	0.291	0.94 (0.44, 2.02)	0.881
NDP in first trimester	0.5 (0.24, 1.04)	0.062	1.07 (0.44, 2.61)	0.877
HCP in first trimester	0.71 (0.36, 1.4)	0.316	1.23 (0.49, 3.07)	0.659
LCP in second trimester	1		1	
SDP in second trimester	1.14 (0.6, 2.16)	0.686	0.52 (0.25, 1.1)	0.088
NDP in second trimester	1.66 (0.79, 3.48)	0.181	0.89 (0.37, 2.17)	0.801
HCP in second trimester	1.67 (0.78, 3.59)	0.191	0.84 (0.35, 2.04)	0.704
CABS				
LCP in first trimester	1		1	
SDP in first trimester	0.88 (0.46, 1.69)	0.704	0.74 (0.32, 1.75)	0.497
NDP in first trimester	0.92 (0.42, 1.99)	0.83	0.79 (0.28, 2.19)	0.644
HCP in first trimester	0.8 (0.37, 1.76)	0.586	0.65 (0.23, 1.87)	0.425
LCP in second trimester	1		1	
SDP in second trimester	0.62 (0.33, 1.18)	0.147	0.97 (0.39, 2.41)	0.941
NDP in second trimester	0.77 (0.35, 1.67)	0.503	1.67 (0.57, 4.86)	0.347
HCP in second trimester	0.8 (0.36, 1.79)	0.585	2.34 (0.83, 6.62)	0.109

Abbreviations: LCP, Low-consumed pattern; SDP, Southern dietary pattern; NDP, Northern dietary pattern, CABS, Clancy Autism Behavior Scale; C-ASQ, Conner’s Abbreviated Symptom Questionnaire. a: Model 2 included covariates such as child age, maternal education, residence, maternal age, pregestational BMI, monthly income, and maternal depression. b: No credible results because there are no children with peer relationship problems among the girls of mothers with HCP in first trimester.

## Data Availability

The datasets analyzed during the current study are available from the corresponding author upon reasonable request.

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
