# Peer review of "Prenatal Healthy Dietary Patterns Are Associated with Reduced Behavioral Problems of Preschool Children in China: A Latent Class Analysis"

_ijerph, 2023, doi:10.3390/ijerph20032214_

Round 1

Reviewer 1 Report

Thank you for inviting me as a reviewer of this valuable manuscript. This study focused on 'Prenatal healthy dietary patterns is associated with reduced neuropsychological problems of preschool children in China: a latent class analysis'. I recommend following suggestions for improving the quality of manuscript.

(Comment 1) Is the China-Anhui Birth Cohort Study (C-ABCS) a national cohort? Are these cohort results to be generalized? If not, I recommend authors to add this point in Discussion Section as a limitation.

(Comment 2) I recommend authors to supplement 'Lterature Review Section'. Summary of previous studies (similar studies) and cohort studies in other countries must be presented in this section

(Comment 3) I recommend authors to add Conclusion Section.

Author Response

Dear Editors and Reviewers:

I quite appreciate your favorite consideration and the reviewer’s insightful comments. Now I have revised the ijerph-2135027 exactly according to the reviewer’s comments, and found these comments are very helpful. I hope this revision can make my paper more acceptable. The revisions were addressed point by point below.

(Comment 1) Is the China-Anhui Birth Cohort Study (C-ABCS) a national cohort? Are these cohort results to be generalized? If not, I recommend authors to add this point in Discussion Section as a limitation.

Response to comment: C-ABCS is not a national cohort and the dietary patterns in the paper are only representative of the diet of people living in central China, which made extrapolation of the results of this study limited. However, our findings are still relevant for generalization. Then Considering the Reviewer’s suggestion, relevant descriptions were added to the research deficiencies section of the paper.

(Comment 2) I recommend authors to supplement 'Literature Review Section'. Summary of previous studies (similar studies) and cohort studies in other countries must be presented in this section.

Response to comment: Based on the reviewers' suggestions, the second paragraph of the discussion was revised as 'Literature Review Section'.

(Comment 3) I recommend authors to add Conclusion Section.

Response to comment: As suggested by the reviewer, the Conclusion Section has been added to the paper.

We appreciate for editors and reviewers’ warm work earnestly, and hope that the correction will meet with approval. Once again, thank you very much for your comments and suggestions.

Reviewer 2 Report

This study highlights the importance of certain nutrients commonly seen in a group of foods and its co-relation to behavioral problems. Authors may want to mention other factors such as birth weight, family history of behavioral problems that are an important factor. Perhaps choice of scales EG BASC a general behavioral scale versus going to Conners and discussing the bias in the choice of scales. Perhaps this discussion may reach a wider audience by adding policy and governance. 

Author Response

Dear Editors and Reviewers:

I quite appreciate your favorite consideration and the reviewer’s insightful comments. Now I have revised the ijerph-2135027 exactly according to the reviewer’s comments, and found these comments are very helpful. I hope this revision can make my paper more acceptable. The revisions were addressed point by point below.

(Comment 1) This study highlights the importance of certain nutrients commonly seen in a group of foods and its co-relation to behavioral problems. Authors may want to mention other factors such as birth weight, family history of behavioral problems that are an important factor. Perhaps choice of scales EG BASC a general behavioral scale versus going to Conners and discussing the bias in the choice of scales. Perhaps this discussion may reach a wider audience by adding policy and governance.

Response to comment: In the process of the analysis, we took into account important factors such as low birth weight and prematurity that affect the behavioral development of the offspring, but their association was not found to be statistically significant. Regarding family history of behavioral problems, one of the exclusion criteria of the cohort was the presence of mental illness and severe neurological disorders in pregnant women and their couples. Specific inclusion and exclusion criteria are described in the team's previously published literature[1].

Childhood follow-up is completed during the routine physical examination of children. In order to save more time for follow-up subjects, children's behavior was assessed by a regular questionnaire (Strengths and Difficulties Questionnaire) and two short questionnaires (Clancy Autism Behavior Scale and Conner's Abbreviated Symptom Questionnaire). Meanwhile, Conner's Abbreviated Symptom Questionnaire may be the most effective diagnostic tool in assessing ADHD because of its brevity and high diagnostic accuracy[2]. However, the children with behavioral problems screened out by the questionnaires are not further diagnosed by clinicians, which is a defect in our design. We will explain this point in the research deficiencies section in the paper.

In addition, it is not easy to add policy and management discussions because diet has cultural attributes and regional characteristics. Even so, we added an appeal at the end of the article to increase the intake of fruits and vegetables for pregnant women.

We appreciate editors and reviewers’ warm work earnestly, and hope that the correction will meet with approval. Once again, thank you very much for your comments and suggestions.

[1] Tao FB, Hao JH, Huang K, Su PY, Cheng DJ, Xing XY, et al. Cohort Profile: the China-Anhui Birth Cohort Study. Int J Epidemiol 2013;42:709-21. doi: 10.1093/ije/dys085

[2] Chang LY, Wang MY, Tsai PS. Diagnostic Accuracy of Rating Scales for Attention-Deficit/Hyperactivity Disorder: A Meta-analysis. Pediatrics. 2016 Mar;137(3):e20152749. doi: 10.1542/peds.2015-2749. 

Reviewer 3 Report

Dear editor,

I have carefully assessed this manuscript. The overall work is of interest and, from my point of view, a series of points should be addressed before the manuscript could be considered for a publication:

- "neurodevelopmental behavior": the use of this term appears somehow generic and not supported by the current clinical literature (eg DSM-5). The authors should adopt a more descriptive and clear terminology

- p values should be reported with a low-case "p" (instead of P)

- "Maternal SDP in early and mid-pregnancy predicted better neuropsychological development of offspring": I think this conclusion is not supported by the results. Assessing the neuropsychological development of an individual require a longitudinal and clinical assessment of his/her history. This terminology and the meaning behind it should be changed across the whole manuscript

- Methods: "using 0-6-year-old pediatric examination table of neuropsychological development": the authors should more clearly report which assessment are they referring to with this sentence

- The explanation of the assessment and the relative timings would benefit the use of a scheme/figure

- Results: The authors should, as a starting point, report a "Table 1" with the main features of the sample

- The authors speak about gender. The use of this word is preferrable when the individuals may actively report which gender the feel they belong to. Otherwise, "sex" may be preferrable

- "data indicated that 121 (7.8%) of children presented with behavioral disorders," behavioral disorders is a non-DSM-5-supported terminology. More descriptive and official terms should be used

- "autistic behavior disorder": same as for the previous point

- The limitations section should discuss the lack of a clinical assessment for ADHD or ASD in the included subjects, while most of the data rely on questionnaires 

Author Response

Dear Editors and Reviewers:

I quite appreciate your favorite consideration and the reviewer’s insightful comments. Now I have revised the ijerph-2135027 exactly according to the reviewer’s comments, and found these comments are very helpful. I hope this revision can make my paper more acceptable. The revisions were addressed point by point below.

(Comment 1) "neurodevelopmental behavior": the use of this term appears somehow generic and not supported by the current clinical literature (eg DSM-5). The authors should adopt a more descriptive and clear terminology

Response to comment: According to the reviewers' comments, we have changed "neurodevelopmental behavior" to "behavioral problems".

(Comment 2) p values should be reported with a low-case "p" (instead of P)

Response to comment: We have made corresponding changes with reference to the suggestions.

(Comment 3) "Maternal SDP in early and mid-pregnancy predicted better neuropsychological development of offspring": I think this conclusion is not supported by the results. Assessing the neuropsychological development of an individual require a longitudinal and clinical assessment of his/her history. This terminology and the meaning behind it should be changed across the whole manuscript

Response to comment: After our discussion, we also found that this description was not appropriate. We changed this sentence to “Maternal SDP in early and mid-pregnancy predicted reduced behavioral problems in preschool children.”, and modified it in other parts of the manuscript.

(Comment 4) Methods: "using 0-6-year-old pediatric examination table of neuropsychological development": the authors should more clearly report which assessment are they referring to with this sentence

Response to comment: 0-6-year-old pediatric examination table of neuropsychological development was used to evaluate the development of language, gross movements, fine movements, adaptability and social behavior of 1-year-old offspring. However, our team did not obtain behavioral assessment data for toddlers, resulting in a lack of continuity in the assessment of child development. Based on the advice of other team members, we should focus on exploring the relationship between diet during pregnancy and behavioral problems in preschool children. This sentence has been deleted.

(Comment 5) The explanation of the assessment and the relative timings would benefit the use of a scheme/figure

Response to comment: A flow diagram was given in the new Fig. 1

(Comment 6) Results: The authors should, as a starting point, report a "Table 1" with the main features of the sample

Response to comment: The main features of the sample was shown in the new Table 1.

(Comment 7) The authors speak about gender. The use of this word is preferrable when the individuals may actively report which gender the feel they belong to. Otherwise, "sex" may be preferrable

Response to comment: We have changed "gender" to "sex" in the manuscript.

(Comment 8) "data indicated that 121 (7.8%) of children presented with behavioral disorders," behavioral disorders is a non-DSM-5-supported terminology. More descriptive and official terms should be used

Response to comment: We have changed "behavioral disorders" to "behavioral problems" in the manuscript.

(Comment 9) "autistic behavior disorder": same as for the previous point

Response to comment: We have changed "behavioral disorder" to "behavioral problem" in the manuscript.

(Comment 10) The limitations section should discuss the lack of a clinical assessment for ADHD or ASD in the included subjects, while most of the data rely on questionnaires

Response to comment: Considering the Reviewer’s suggestion, we have relevant limitations, and the fourth point of the limitations section is described in detail.

We appreciate editors and reviewers’ warm work earnestly, and hope that the correction will meet with approval. Once again, thank you very much for your comments and suggestions.

Round 2

Reviewer 3 Report

Dear Editor, 

My points have been addressed

Best regards

Jacopo Pruccoli